# Feasibility, Safety, and Satisfaction of Combined Hysterectomy with Bilateral Salpingo-Oophorectomy and Chest Surgery in Transgender and Gender Non-Conforming Individuals

**DOI:** 10.3390/ijerph18137133

**Published:** 2021-07-03

**Authors:** Ilaria Mancini, Davide Tarditi, Giulia Gava, Stefania Alvisi, Luca Contu, Paolo Giovanni Morselli, Giulia Giacomelli, Alessandra Lami, Renato Seracchioli, Maria Cristina Meriggiola

**Affiliations:** 1Dipartimento di Scienze Mediche e Chirurgiche, IRCCS Azienda Ospedaliero-Universitaria di Bologna University of Bologna, 40138 Bologna, Italy; davide.tarditi@hotmail.it (D.T.); giulia.gava2@unibo.it (G.G.); stefania.alvisi@gmail.com (S.A.); luca.contu1989@gmail.com (L.C.); paologiovanni.morselli@unibo.it (P.G.M.); giuliagiacomelli90@gmail.com (G.G.); lami.alessandra@gmail.com (A.L.); renato.seracchioli@unibo.it (R.S.); cristina.meriggiola@unibo.it (M.C.M.); 2Gynecology and Physiopathology of Human Reproduction Department of Medical and Surgical Sciences (DIMEC), University of Bologna, 40138 Bologna, Italy; 3Plastic Reconstructive and Aesthetic Surgery Department S. Orsola–Malpighi Hospital, Alma Mater Studiorum–University of Bologna, 40138 Bologna, Italy

**Keywords:** transgender, gender affirming surgery, chest surgery, transmen

## Abstract

The demand for masculinizing breast surgery and hysterectomy with bilateral salpingo-oophorectomy (HBSO) from transmen has increased. With a multidisciplinary approach, these surgeries can be performed in a single session. The objective of this study was to retrospectively evaluate the feasibility, safety, and satisfaction of HBSO and chest surgery in transmen. A cohort of 142 subjects who underwent HBSO alone or combined with chest surgery at Sant’Orsola Hospital was analyzed. Intra and post operation events were evaluated. Subjective post-intervention satisfaction, acceptability, and impact of intervention were assessed via a semi-structured interview. Nineteen transmen underwent HBSO alone and 123 underwent combined surgery. HBSO was performed laparoscopically in 96.5% of transmen (137/142). As expected, length of hospital stay and blood loss were significantly higher in the combined surgery group. A total of 13 intra or post-operative complications occurred in the combined surgery group (10.5%) with thoracic hematoma being the most frequent complication (7.6%). Only one rare complication occurred in the HBSO group (omental herniation through a laparoscopic breach). The overall subjective satisfaction score was 9.9 out of 10 for both groups. Positive changes in all areas of life were reported, with no significant differences. We found that the combined surgery appears to be well tolerated, safe, and feasible in transmen and satisfaction with the combined procedure was high in all subjects.

## 1. Introduction

The demand for gender affirming surgery (GAS) has increased substantially over the last decade and recent data have reaffirmed the positive role of GAS on mental and psychosocial health [1,2]. Chest masculinization and removal of female organs are often required by transgender men and gender non-conforming individuals as a first step of gender affirming surgery. All routes of hysterectomy are feasible and safe when performed for gender affirmation, with no documented increased risk of complications compared to the cisgender population and are associated with an improved quality of life [3,4,5,6]. Minimally invasive approaches are the most common (laparoscopy, vaginal, laparoscopic-assisted vaginal, and robot-assisted laparoscopic route), each with particular benefits for transmen in terms of scarring, hospitalization time, and conservation of vasculature for potential future gender affirming surgeries [3,7,8].

Combining chest masculinization and hysterectomy with bilateral salpingo-oophorectomy (HBSO) in the same operation may carry some advantages such as lower level of anxiety and stress and lower cost. Four studies have been published on this topic and outcomes on less than 400 combined surgeries have been reported [3,7,8,9]. However, there is no agreement as to whether the combined surgery is associated with a higher complication rate when compared to the single surgery. In studies that have described combined surgery, the complication rates range from 7.4 to 25.3%. Elfering et al. reported an increased risk of complications after combined surgery compared with separately performed chest surgery, while other authors did not find significant differences in the complication rate when comparing the combined procedure to chest masculinizing surgery alone [3,7,8,9]. It is not clear whether the order of surgeries has an influence on the complication rate. Ott hypothesized that performing the mastectomy first may cause additional hematoma complications, however the same group revoked their conclusion in a more recent study [9]. Importantly, no studies have reported on acceptability, tolerability, and satisfaction of these surgeries in the trans population.

The aim of this study was to evaluate the feasibility, safety, acceptability, and satisfaction of combined surgery or HBSO alone in transgender men.

## 2. Materials and Methods

Charts from a cohort of people (*n* = 142) diagnosed with gender dysphoria according to DSM-V [10] were retrospectively reviewed. The aim was to determine if there was a difference in outcome and satisfaction between those subjects who had undergone HBSO associated or not with chest surgery at the Gynecology Clinic of S. Orsola Hospital in Bologna between April 2006 and December 2020.The combined surgery was carried out by the simultaneous work of a team of gynecologists and plastic surgeons while the HBSO was carried out by the gynecological team.

This study was approved by the Medical Ethics Committee of S. Orsola Hospital, Bologna, and all subjects gave written informed consent.

According to the Endocrine Society in 2017 and WPATH guidelines in 2012, GAS was performed after at least one year of hormonal therapy and in transgender individuals over 18 years which is the legal age in Italy [11,12]. All transgender people who underwent GAS before the publication of the 2012 WPATH guidelines, completed the so-called “Real Life Experience phase” and obtained legal approval (as is required by Italian law) prior to GAS.

Data collection derived from the retrospective consultation of medical records.

Anthropometric measurements were taken from all subjects: stature was measured with a stadiometer as the distance from the vertex to the floor by asking the subject to stand erect, barefoot with their shoulders touching the wall. Body mass index (BMI) was calculated as weight in kilograms divided by the square of height in meters (kg/m^2^). Nationality was obtained from identity documents required at hospital admission, while occupation and the other clinical information were collected by consulting patient medical records. During the pre-operative assessment, a transabdominal pelvic ultrasound was performed in all subjects, for the evaluation of pelvic anatomy [13].

Anamnestic data regarding the surgical procedure and clinical and biochemical parameters during the pre- and post-operative course were collected and analyzed.

Intra-operative data included total surgery time, partial times of HBSO and chest masculinization, technique used for gynecological surgery (laparotomic and laparoscopic), plastic surgery techniques, and anesthesiological risk assessed according to the ASA (American Society of Anesthesiology) score [14].

Post-operative hospitalization data included length of hospital stay, hemoglobin (Hb) and hematocrit (Ht) values measured before surgery and on the first post-operative day, intensity of post-operative pain quantified according to the visual analogic scale (VAS), and type of analgesia.

The visual analog scale (VAS) was used to measure pain intensity during post-operative hospitalization [15]. Scores were recorded by making a handwritten mark on a 10 cm line representing a continuum between “no-pain” and “worst-pain” [16].

In addition, both intra- and post-operative complications related to the procedure were recorded, such as hematoma formation, intraperitoneal blood collection, significant genital blood loss, need for blood transfusion, need for re-intervention during hospitalization, and need for rehospitalization following discharge.

Satisfaction of individuals following surgery was investigated by interviews performed one year after surgery assessed by numerical rating scale (NRS) with scores from 0 to 10. Questions were designed to investigate: the subjective need for performing the intervention; subjective positive changes in social, family, work, and sexual domains; whether both interventions brought about significant changes equally or whether one was more significant than the other; satisfaction with performing both interventions in a single session; ease of access to care (see Appendix A).

### 2.1. Surgical Techniques

Chest masculinization surgery was performed according to the most appropriate technique, chosen on the basis of the size and degree of breast ptosis and on skin elasticity. Techniques used were:Pull-through technique: in case of small or medium breast volume with good skin elasticity.Concentric circular technique: for small/medium-sized breasts with poor skin elasticity.Vertical bipedicle technique: in case of moderate-sized breast with poor skin elasticity or large breast with good skin elasticity.Glandular resection with free nipple grafting: for ptotic and large breasts with poor elasticity and female appearance [17].

The HBSO procedure was performed laparoscopically in the majority of cases. Where necessary, the approach was laparotomic with a transverse incision according to the Pfannenstiel technique or, in one case, longitudinal sub-umbilical-pubic incision.

When subjects underwent concurrent clitoris surgery, a simple metoidioplasty was performed [18].

This surgery is performed on the clitoris enlarged by pre-operative use of testosterone therapy; this surgical procedure includes a subcoronal skin incision followed by degloving and transection of clitoris suspensory ligaments [18]. As described by Djordevic et al. [18], simple metoidioplasty does not include urethra lengthening.

### 2.2. Statistics

All continuous data are expressed as mean and standard deviation of the mean and all categorical data was expressed by frequency rate and percentage. The Kolmogorov–Smirnov test was used to assess the normality of distributions. When data were normally distributed, the parametric Student’s t-test was used to assess differences between two groups. Otherwise, the Mann–Whitney test was used. Fisher’s nonparametric chi-squared test was performed to investigate the relationships between categoric variables. Statistical analysis was carried out by means of the Statistical Package for the Social Sciences (SPSS) software version 23.0 (International Business Machines Corp., Armonk, NY, USA).

## 3. Results

A total of 142 transmen who had undergone gender affirming surgery were included in the study. All subjects underwent HBSO with 123 of these having masculinizing breast surgery in the same surgical session.

### 3.1. Demographic and Basal Characteristics

Demographic and basal characteristics of the two groups are shown in Table 1. No significant differences in major demographic characteristics were detected between the two groups, with the exception of longer T assumption in the HBSO alone group (2.8 ± 1.4 vs. 4.5 ± 3.5 years, *p* < 0.05). A higher but not statistically significant rate of current smoker was detected in the HBSO alone group (43.9 vs. 63.2%).

Pre-operative blood tests, performed four weeks before surgery, did not show significant differences between the two groups.

In 123 of 142 subjects, no pathologic ultrasonographic findings were found. Uterine myomas were identified in 17 transgender men, which in four cases were voluminous (>9 cm). Two cases of endometrial polyps and two endometriosis ovarian cysts were found during the pre-operative pelvic ultrasound.

### 3.2. Surgical Procedures

Table 2 shows data on surgical procedures and outcomes in the two groups. Nineteen subjects underwent total laparoscopic HBSO alone; in one subject a simple metoidioplasty was performed concurrently. One hundred and twenty-three subjects underwent combined surgery with five undergoing laparotomic HBSO, with the remaining 118 operated laparoscopically; of the latter, 10 underwent concurrent simple metoidioplasty. Laparotomy was performed in four out of five cases because of the presence of voluminous uterine myomas, with the fifth case being performed for anesthesiologic reasons.

All received antibiotic prophylaxis in the operating room with first- or second-generation cephalosporins administered intravenously. Appropriate antithrombotic prophylaxis was also set based on thrombotic risk, assessed by Caprini score [19].

Among the different chest masculinization techniques, the most frequently used technique was glandular resection with free nipple grafting (61.8%, 76 cases), followed by the pull-through technique (23.6%, 29 cases), the vertical bipedicle technique (13.0%, 16 cases), and the concentric circular technique (1.6%, 2 cases).

In 36 out of 123 subjects, gynecological surgery was the first procedure performed whereas plastic surgery was performed first in the remaining 87 cases. The order of the surgeries was set on the basis of surgeon availability.

### 3.3. Post-Operative Data

Post-operative data are summarized in Table 3. Blood tests performed on the first post-operative day revealed a statistically significant reduction in hemoglobin and hematocrit compared to the pre-operative day in both groups. Comparing those who had undergone gynecological surgery first and those who underwent chest surgery first, there was no significant difference in the drop of blood parameters investigated (Hb: −3.8 ± 0.2 g/dL vs. −3.9 ± 0.3 g/dL, *p* > 0.05; Ht: −11.9 ± 5.5% vs. −11.9 ± 3.4% HBSO first, *p* > 0.05 in the chest first (*n* = 82) and HBSO first (*n* = 36) group, respectively).

Opioid-based analgesia in continuous infusion via an elastomeric pump or through a patient controlled analgesia (PCA) system was set for most subjects for the first 48 h after combined surgery, possibly associated with administration of nonsteroidal anti-inflammatory drugs at fixed times.

In the HBSO alone group, 12 of 19 individuals (63.2%) used opioids, administered in only seven (36.8%) of these by continuous infusion. Conversely, in 121 out of 123 (98.4%) subjects who underwent combined surgery, opioids were used, mostly morphine, administered by elastomeric pump for 48 h. Opioid administration was statistically significant between the two groups (*p* < 0.01), with a lower dose used in the HBSO alone group.

Post-operative pain mean scores reported up to the third post-operative day are shown in Table 3. No significant differences were found between the two groups.

### 3.4. Complications

Table 4 shows the main complications that occurred during and following surgery. The difference between the complication rates of the combined surgery and HBSO alone group was not significant, despite the percentage of complication being higher in the combined surgery group than in the single intervention group (10.5% combined surgery vs. 5.3% HBSO alone surgery, *p* > 0.05). No minor complications have been reported in the laparoscopic combined or HBSO alone surgery, while one has been reported in a subject who underwent combined surgery via the laparotomic route. In this subject, hypoparesthesia of the right lower limb was reported, caused by the surgical retractor used during the laparotomic HBSO which resolved spontaneously.

An intraoperative complication occurred in one case: difficulties were encountered during intubation leading to significant airway edema and the need for post-operative monitoring in the intensive care unit. The same subject underwent laparotomy given the impossibility of obtaining an adequate pneumoperitoneum for anesthesiological reasons and subsequently underwent further surgery for abdominal dehiscence.

During post-operative hospitalization, hematomas of the thoracic region occurred in nine cases, three of which underwent immediate drainage. Eight out of nine thoracic hematomas observed after combined surgery occurred when chest masculinizing surgery was performed first and only one where the subject underwent gynecological surgery first (1/36, 2.8%); however, this difference is not statistically significant (*p* = 0.28). Blood transfusions were performed on the third or fourth post-operative day in seven individuals (4.9%), all of whom had undergone combined surgery. Six out of seven (86%) subjects who required blood transfusions had a post-operative time complicated by thoracic hematoma, whereas only one transfusion was required because of the abdominal hemoperitoneum.

In two cases it was necessary to proceed to immediate re-intervention: in one case a laparoscopic abdominal toilette was performed because of the hemoperitoneum while in another case repositioning of omental herniation through a laparoscopic breach was performed.

Later complications included dehiscence, found after laparotomy, as already described above, which was repaired with a biological prosthesis six months after surgery.

Overall, the analysis found one complication in the HBSO group (5.3%), while the patients with one and more than two complications in the combined-surgery group were 12 (9.8%) and 1 (0.8%), respectively. No significant differences have been detected between these groups (*p* value = n.s).

Twenty-eight individuals (22.8%) required additional procedures for esthetic improvement of the breast such as liposuction, lipofilling, and revision of surgical scars or of the nipple-areola complex at least six months later.

### 3.5. Satisfaction

Seventy-four (52.1%) individuals responded to the questionnaire during follow up (see Appendix A). Of these 74 patients, 63 had undergone combined surgery (51.2%), whereas 11 had undergone HBSO alone (57.9%). The remainder of the subjects either did not come to the follow-up visits or refused to respond to the questionnaire.

Table 5 summarizes data regarding importance, preference, and impact of surgery on different aspects of life in the two groups.

Performance of the surgical intervention was considered important by all subjects (9.9 ± 0.3 in both groups range from 8 to 10 in combined group and from 9 to 10 in the HBSO alone group). Positive changes in social, family, sexual, and work areas of life were reported in all domains investigated, with no significant differences between the two groups.

Most individuals in both groups defined the chest and genital surgery to be of equal importance, whereas only a few subjects (4.8% in the combined group and 0% in HBSO alone group) attributed greater importance to HBSO.

With regard to satisfaction with the performance of the combined surgery in a single session, the mean score was 9.8 ± 0.7 (range from 7 to 10), whereas 55% of subjects who underwent HBSO alone indicated they would have preferred to undergo the two procedures separately.

Finally, the subjects interviewed quantified the ease of access to the surgery (finding the center, performing pre-surgical examinations and interviews) with a mean value of 8.5 out of 10 in each group (8.5 ± 2.1 in combined group and 8.5 ± 1.0 in HBSO alone group, *p* = n.s).

## 4. Discussion

In this observational retrospective study, the feasibility, safety, and satisfaction of chest surgery and HBSO combined or HBSO alone in transgender men were evaluated. In this cohort the overall complication rate was low with no significant differences between subjects who had undergone combined surgery compared to those who underwent HBSO alone, despite the fact that the percentage of complications was higher in the combined surgery group. Subjects interviewed during follow-up outpatient visits reported high satisfaction with the combined surgery, with tolerable physical commitment.

Gender affirming surgeries represent an important part of treatment of gender dysphoria and high satisfaction levels after GAS in the transgender population have been reported [6,20,21,22]. In 2018 Van de Grift et al. reported high surgical satisfaction rates despite a considerable number of post-operative complications in a cohort of 136 transgender men and women who had undergone single or multiple GAS (vaginoplasty, breast augmentation, facial surgery, vocal cord surgery, mastectomy, hysterectomy, phalloplasty, metoidioplasty) [6]. This finding emphasized the positive impact of gender affirming procedures on the wellbeing and quality of life of transgender individuals. In our cohort, all transgender men perceived surgical interventions of GAS (genital and/or chest surgery) as very important for their wellbeing, with positive subjective changes in different areas of their life with no differences between the two groups. Although many authors have underlined the importance of chest surgery for transmen, these data suggest that HBSO is also important and has an impact on the life of trans men [17,22]. Confirming this, 54 to 63% of the transmen in our groups reported both interventions to be equally important. These results are unexpected but can be explained by the necessity of gender affirming genital surgery as a prerequisite for identity document changes according to Italian law until 2015.

In this cohort, the subjects were young with normal BMI, few co-morbidities, and low anesthesiological risk. Almost all studies that have analyzed combined surgery in transmen reported demographic and basal characteristics similar to this sample and this facilitated the comparison of results [3,8,9,23]. Performing two surgical interventions in the same operation session is advantageous but may potentially carry increased risks and complications. In our series, the overall complication rate was as low as 9.9%. In the group undergoing combined surgery the complication rate was 10.5% (13/123 subjects) whereas it was 5.3% (1/19 subjects) in the HBSO alone group; however, due to the small numbers of the HBSO alone group the comparison does not achieve significance. Nine complications of the combined surgery group were due to chest surgery (7.6%) and four (3.2%) to the HBSO surgery. This overall rate of complications is similar to that reported in the literature (complication rates ranging from 5.4% to 25.3%) [3,7,9,23]. One of the most frequent complications, concluded by the analysis of both combined and chest masculinization-only procedures, is thoracic hematoma [24]. A recent study by Elfering et al. compared the combined reduction mammoplasty and laparoscopic HBSO with reduction mammoplasty alone (212 cases vs. 268 cases), finding a significant difference in the occurrence of thoracic hematomas and the need for re-intervention between the two groups (16% vs. 7.5%, respectively). The order of surgeries did not affect the complication rate (16.4% vs. 15% in the HBSO first versus mastectomy first group, respectively). The frequency of thoracic hematoma in this sample of combined surgery was 7.6%, which is lower than that described by Elfering and previous reports of combined surgery (12–17%) [7,9]. The analysis did not detect any significant difference in the complication rate according to the order of surgeries (9.7% in chest surgery first vs. 2.8% HBSO surgery first *p* = n.s.).

A higher complication rate in the HBSO performed during the combined surgery compared to HBSO performed alone was reported, however three out of four complications occurred with the laparotomic route which is comparable to the complication rate of HBSO in the cisgender population (4.5%) [25].

Bretschneider et al. analyzed the complication rate of HBSO alone surgery performed with the use of minimally invasive techniques (both vaginal and laparoscopic) in transgender and gender non-conforming individuals and found post-operative complications in 3.1% of cases, which is similar to our results. With regard to gynecological surgery alone, only one major complication occurred in this sample, which was omental herniation that required immediate re-intervention. Trocar hernias represent a rare complication of abdominal laparoscopic surgery, with frequency significantly associated with the increasing size of the trocars applied (from 0.02% to 5% in >10 mm trocars) [26].

As expected, surgery time, length of hospital stay, and blood loss were significantly higher in the combined procedure group compared to the HBSO alone group. The mean hospital stay length for combined surgery was 4.4 days with 2.7 days for HBSO alone, in line with the hospital stay for this type of surgery [27,28]. Data obtained from similar studies have indicated mean hospitalization stay ranging from zero to eight days [8,9,17,20]. Cizek et al. recently highlighted the possibility of discharge on the same day of surgery and therefore the feasibility of the combined surgery of HBSO and bilateral mastectomy as day surgery [8]. The small number of subjects examined however, does not guarantee significant reliability, and further studies should be undertaken to confirm this possibility. In our clinic, however, the duration of hospital stay has also progressively decreased over the years.

Testosterone therapy remains a confounding factor in estimating blood loss because pre-operative blood tests are performed at least one month before surgery, at which time hormone therapy administration is temporarily discontinued in preparation for surgery in our center. It is therefore unclear if or by how much the decrease in testosterone contributes to the decrease in hemoglobin concentration, generating an overestimation of intraoperative bleeding. However, the need for testosterone discontinuation before surgery is still unclear: the possible increase of venous thromboembolic risk and intra- and post-operative bleeding as well as an increased risk of chest hematomas, have been reported [9,29,30,31]. In 2017, Cizek et al. reported data regarding combined surgery in transgender patients who did not stop taking testosterone prior to surgery, showing no difference in the intra- and post-operative complication rate [8]. Similar results were published by Berry et al. in 2011 [32]. Considering the results of these studies, our hospital has recently changed its practice and no longer require the cessation of testosterone before surgery.

In the available literature, there is no data regarding the assessment and management of post-operative pain after GAS. From this analysis, using the VAS scale of pain assessment, on the first post-operative day, the average score was 2.9 and, surprisingly, it was the same in those subjects who had undergone HBSO alone surgery and in those who underwent combined surgery. This data is in line with post-operative pain in the cisgender population undergoing laparoscopic HBSO for benign reasons [33]. Overall post-operative pain was adequately monitored and controlled with therapy, with no occurrence of major side effects.

Combined surgery seems to be feasible, safe, and tolerable in transmen and gender non-conforming individuals, with clear advantages and high surgical satisfaction rates. The reduced number of sessions may decrease the overall psychological stress perceived during the gender affirming surgical procedure. It is well known that individuals facing surgery develop high levels of stress and anxiety [34,35], with effects on both subjective wellbeing and the post-operative healing process [36,37,38]. Combined surgery has a longer duration, involves more severe anemia, and comes with a greater number of days of hospitalization when compared to single procedures. This may increase the risk of both intra- and post-operative complications [39]. In line with other studies, transgender men undergoing surgery are often young individuals with no major comorbidities and low anesthesiologic risk, able to tolerate surgeries of a longer duration [3,8,9]. In addition, intra- and post-operative complications do not seem to be related to the combination of the two procedures in a single session, but rather to the individual procedures performed [3]. These data confirm that the combined surgery results in high satisfaction of the subjects and is well tolerated.

Although the response rate was around 50%, mainly due to the loss of subjects during follow up, those who responded reported excellent satisfaction with the execution of the combined procedures in a single session (9.9 out of 10). The interviewed individuals assigned an average score of 5.0 to the physical effort perceived in terms of pain and post-operative recovery for the combined group and 3.5 for the HBSO alone group, thus confirming, the tolerability of the post-operative course of both the procedures, with no significant differences.

Some limitations of this study should be acknowledged. The main limitation is the retrospective design, which can cause data to go missing during extraction and is prone to bias. The data was collected over fourteen years (2006–2020). During these years, the evolution of surgery has been significant, and this could be a confounding factor in this analysis. The small number of people in the HBSO alone group makes the direct comparison with the combined surgery group difficult. Reporting of data within the center for a cohort that had chest surgery alone would have provided further information, but these data are not available. The study was conducted only in our university hospital center in which residents are involved in the surgery therefor the length of surgery may have been increased by the teaching time. A semi-structured interview to evaluate satisfaction, physical effort, and changes in life after surgery were performed while validated questionnaires may have provided additional and more reliable data. Additionally, many subjects were lost to follow up and the response rate to the semi-structured interview was low.

## 5. Conclusions

In conclusion, this data highlights the safety and feasibility of the combined surgery of HBSO and chest masculinization in transgender men with a low complication rate. These data regarding subjective satisfaction and advantages associated with single session surgery support the performance of this combined surgical procedure in transgender men and gender non-conforming individuals.

## Figures and Tables

**Table 1 ijerph-18-07133-t001:** Demographic and basal characteristics of subjects undergoing combined surgery and hysterectomy with bilateral salpingo-oophorectomy (HBSO) alone.

Characteristics	Combined Surgery (*n* = 123)	HBSO Alone (*n* = 19)
Age (years; mean ± SD)	32.2 ± 6.7 (21–51)	29.9 ± 63 (22–50)
Nationality		
Italian	111 (90.2)	17 (89.5)
Other ^	12 (9.8)	2 (10.5)
Occupation (%)		
Employed	97 (78.8)	11 (57.9)
Student	6 (4.9)	2 (10.5)
Unemployed	20 (16.3)	6 (31.6)
BMI (kg/m^2^)	24.50 ± 4.08	23.37 ± 3.94
underweight	2 (1.6)	1 (5.3)
normal weight	76 (61.8)	13 (68.4)
Overweight	33 (26.8)	4 (21.0)
obese I	10 (8.1)	1 (5.3)
obese II	1 (0.8)	0
obese III	1 (0.8)	0
Current smoker	54 (43.9)	12 (63.2)
GAHT duration	2.8 ± 1.4	4.5 ± 3.5 *
Comorbidity ᶧ	18 (14.6)	5 (26.3)
Pre-operative blood tests		
Hb (g/dL)	15.5 ± 1.2	15.6 ± 1.3
Ht (%)	45.7 ± 3.6	45.2 ± 3.2
ASA score		
I	52 (42.3)	4 (21.1)
II	70 (56.9)	15(78.9)
III	1 (0.8)	0

Data are reported as mean ± SD or number of subjects (%); BMI: body mass index. GAHT: gender affirming hormonal treatment. ASA: American Society of Anesthesiology; ^ Other includes: Germany (2), Romania (2), Chile (2), Moldavia (1), Switzerland (1), El Salvador (1), Croatia (1), Poland (1), Colombia (1), Perù (1); ᶧ comorbidity: arterial hypertension, thrombophilia, hypothyroidism, clotting factor deficiency, cardiac valvular disease; * *p* < 0.05 vs. combined surgery.

**Table 2 ijerph-18-07133-t002:** Number of subjects undergoing different surgical procedures and number of revisions of chest surgery.

Combined Surgery (*n* = 123)	N° of Subjects (%)
Laparoscopic HBSO + chest masculinization With concurrent simple metoidioplasy	118 (83.1) 10
Laparotomic HBSO + chest masculinization	5 (3.5)
**HBSO alone (*n* = 19)**	19 (100)
Laparoscopic With concurrent simple metoidioplasy	1
**Plastic surgery procedures (*n* = 123)**	
Pull-through technique	29 (23.6)
Concentric circular technique	2 (1.6)
Vertical bipedicle technique	16 (13.0)
Free nipple grafting technique	76 (61.8)
**Revisions of chest surgery ^**	28 (22.8)

Data are reported as mean ± SD or number of subjects (%); ^ revision of chest surgery at least six months later: liposuction, lipofilling, revision of surgical scars or of the areola-nipple complex.

**Table 3 ijerph-18-07133-t003:** Data on postoperative hospital stay, blood loss, and pain.

Post-Operative Data	Total (*n* = 142)	Combined Surgery (*n* = 123)	HBSO Alone (*n* = 19)
*Length of hospital stay (days)*	4.1 ± 2.5	4.4 ± 2.6	2.7 ± 0.6 *
*Blood tests*			
Hb (g/dL) pre-operative	15.5 ± 1.2	15.5 ± 1.2	15.6 ± 1.3
Hb (g/dL) day 1	11.9 ± 1.6	11.7 ± 1.5 #	13.5 ± 1.3 * #
Ht (%) pre-operative	45.7 ± 3.5	45.7 ± 3.6	45.2 ± 3.2
Ht (%) day 1	34.7 ± 4.7	34.1 ± 4.5 #	38.9 ± 3.9 * #
Change from baseline Hb (g/dL)	−3.6 ± 1.4	−3.8 ± 1.2	- 2.1 ± 1.2 *
Change from baseline Ht (%)	−10.9 ± 4.1	−11.6 ± 3.7	−6.4 ± 3.7 *
*Post-operative pain (VAS 10)*			
Day of surgery	2.82 ± 1.65	2.87 ± 1.66	2.50 ± 1.63
Day 1	2.81 ± 1.74	2.90 ± 1.71	2.25 ± 1.98
Day 2	2.09 ± 1.47	2.22 ± 1.47	1.23 ± 1.17 §
Day 3	1.57 ± 1.09	1.63 ± 1.09	1.00 ± 1.00

Data are reported as mean ± SD or number of subjects (%); Hb: hemoglobin; Ht: hematocrit; VAS: visual analogic scale; NSAD: nonsteroidal anti-inflammatory drugs; * *p* < 0.001 vs. combined surgery, § *p* < 0.05 vs. combined surgery, # *p* < 0.001 vs. preoperative values.

**Table 4 ijerph-18-07133-t004:** Complications during and after surgeries (*n* (%)).

Complications			Total (*n* = 142)	Combined Surgery (*n* = 123)	HBSO Alone (*n* = 19)
*Laparoscopic route*			*n = 137*	*n = 118*	*n = 19*
	Major complications	Thoracic hematomas	9 (6.5)	9 (7.6)	0
		Hemoperitoneum with reintervention	1 (0.7)	1 (0.8)	0
		Omental herniation with reintervention	1 (0.7)	0	1 (5.3)
*Laparotomic route*	Minor complications		0 *n = 5*	0 *n = 5*	0 *n = 0*
	Major complications	Dehiscence with reintervention	1 (20)	1 (20)	0
		Femoral nerve paresthesia	1 (20)	1 (20)	0
	Minor complications	Anesthesiologic complications	1 (20)	1 (20)	0
*Total complications*			14 (9.9)	13 (10.5)	1 (5.3)

Data are reported as number of subjects (%); * *p* < 0.05 vs. combined surgery.

**Table 5 ijerph-18-07133-t005:** Responses to the semi-structured interview regarding importance and impact of the intervention (score range from 0 = no change/importance to 10 = maximum change/importance).

Interview	Combined Surgery (*n* = 63)	HBSO Alone (*n* = 11)	*p* Value
Importance of the surgical intervention	9.9 ± 0.3	9.9 ± 0.3	1.00
Changes in social life	9.3 ± 1.7	9.0 ± 1.2	0.58
Changes in family life	8.6 ± 2.7	7.4 ± 3.2	0.19
Changes in work area	8.7 ± 2.9	8.8 ± 1.6	0.92
Changes in sexual life	9.2 ± 1.8	8.5 ± 2.7	0.28
Physical effort perceived after surgery)	5.0 ± 3.0	3.5 ± 2.6	0.12
Quantify the ease of access to the surgery (out of 10)	8.5 ± 2.1	8.5 ± 1.0	1.00
Greater importance to: n subjects (%)			
- chest surgery	25/63 (39.6%)	4/11 (36.3%)	1.00
- HBSO	3/63 (4.8%)	0/11 (0%)	1.00
- Both equally	34/63 (54%)	7/11 (63.6%)	0.74
- Other surgery	1/63 (1.6%)	0/11 (0%)	1.00

Data are reported as mean ± SD or number of subjects (%).

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
