# Peer review of "Feasibility, Safety, and Satisfaction of Combined Hysterectomy with Bilateral Salpingo-Oophorectomy and Chest Surgery in Transgender and Gender Non-Conforming Individuals"

_ijerph, 2021, doi:10.3390/ijerph18137133_

Round 1

Reviewer 1 Report

the topic has been addressed previously but is still of interest.

A surprising result of this study is that mastectomy and hysterectomy/adnectomy are regarded equally important for this distinct group of patients. That is not in congruence with the literature and the authors should shed more light on this finding. For transgender males breasts are strongly connotated female and hinder recognition as being male. Genital organs of AMAB can be worrying but do not interact directly. Arguments for this finding should be discussed more profoundly.

Regarding data reliability the authors should consider to exclude patients undergoing laparotomy for analysis as only laparoscopic hysterectomy / adnectomy  with or without the combination of mastectomy is the primary endpoint.  Additionally it is well-known that laparotomy causes significantly more pain postoperatively compared to either laparoscopy or the vaginal route of access. Inclusion of this subgroup may have a distorting effect on the results

Complication should be divided in major and minor complication . See also: Ann Surg. 2004 Aug; 240(2): 205–213.

Some points of the manuscript with further questions or in need of correction:

  • on the one hand the authors state that "According to the Endocrine Society in 2017 and WPATH guidelines in 2012, GAS was performed" but only one sentence later it is described that "all transgenders successfully completed the so-called “Real Life Experience phase”. Version 7 of the SoC of WPATH deleted the Real Life Experience phase without substitution ! This condition is not according the SoC in power and it should be commented why this requirement still is a prerequisite in Italy
  • Hospital stay: this item is of limited interest as hospital stays are strongly dependent from national policies rather than medical criteria.
  • Satisfaction following surgery: how long after surgery ? It is well   known, that the results from breast sugery should only be judged after 6 months or later.
  • "simple metadoioplasty" does not exist:  with urethralengthening nothing is simple and complication rates very high. 
  • Preoperative abominal ultrasound: this is NOT a validated method for the judgement of pelvic organs ! Please comment.
  • Table 1. age appears significantly higher in this study compared to the  average age of transmales undergoing surgery in other studies. Please comment. 
  • "antithrombotic prophylaxis (was also set) based on thrombotic risk". Please exemplify how you judge the thrombotic risk.
  • Duration: what are your measurement points ? First incision to last suture ? Time of reposition is included. That makes this item not meaningful and should be excluded
  • Contradiction: in abstract only one major pelvic complication is mentioned, whereas in the text (sentence 221) two complications !
  • "Laparocele". I am not familiar with this term. Do you mean 'dehiscence' 
  • Reconstruction of the nipple-areola complex: this procedure is frequently done in a second step: Please comment why you perform this step in one procedure . Is it thinkable that this is a reason for complications ? 

Reviewer 2 Report

The paper is very interesting, related to a very actual topic, with novelty in the focus. There are a few comments and proposals to improve the quality of this paper:

  • A global suggestion it to write in third-person, there are some references to first-person (We...) and in my opinion, for a scientific paper it sounds better third person.
  • Introduction is very brief and there is information about the topic included in the methods. Maybe it would be more correct to include lines 68-72 and point 2.1. Surgical Techniques into the Introduction.
  • About the Methods, there is no specification about validity or feasibility of data, if the tools for data collection are standard and validated surveys, if the tool includes qualitative data and they have been analyzed in any way and if there are any other variables (such as psychosocial or structural) with possible influences in the variables of the study.
  • In the part Data, there are several characteristics in Table 1 not explained in Methods (lines 144-145).
  • About Table 4, there has not been any kind of correlation analysis to detect and explain if 2 or more complications were connected with the procedures. In my opinion, this is the most interesting part of the study and it could be wonderful if these deep data can be integrated in the paper.
  • Discussion is correct.
  • In Limitations of the study, data has been collected between 2006-2020. During these years, the evolution of surgery and lifestyle have been significative and their influence maybe need to be considered as strange variables in the data analysis, or at less as limitations of the study.
  • Conclusions are correct and connected with the results.

Reviewer 3 Report

A cohort of transmasculine people (n=142) were retrospectively reviewed to determine if there was a difference in outcome and satisfaction between those that had HBSO +/- chest surgery at a single centre.

The majority (n=123) had combined surgery. The small number of people in the HBSO single surgery group makes direct comparison difficult. Ideally, reporting of data within the centre for a cohort that had chest surgery alone would make a useful second control. However, this paper does add to the literature as only 400 combined surgeries have been reported previously.

It is unclear whether these are 142 consecutive surgeries or if some patients were excluded e.g. due to missing data. The authors could comment on whether there was bias in determining who was suitable for the combined surgery, were the ASA scores different using a categorical statistical test?

There is a typographical error in Table 4 – should read ‘Thoracic hematomas’.

Round 2

Reviewer 1 Report

As only few persons are familiar with the term 'simple metadoioplsty' I strongly recommend to add to sentence 168 (simple metadoioplasty'):  ' as described by Djordevic et al (reference ...)  does not include urethra lengthening    
